# Relationship between Body Schema and Scholar Maturity: A Study from the National College of Banat in Timisoara, Romania

**DOI:** 10.3390/children9091369

**Published:** 2022-09-09

**Authors:** Mariana Cristina Șunei, Simona Petracovschi, Eugen Bota, Bogdan Almajan-Guță, Adrian Nagel

**Affiliations:** Faculty of Physical Education and Sport, West University of Timisoara, 300223 Timisoara, Romania

**Keywords:** physical education, preparatory class, body schema, maturity, drawing

## Abstract

The purpose of this study is to analyze the role and importance of specific physical exercises in the development of the body schema in preparatory class pupils and its effect on the development of maturity. Methods: The experiment took place over a period of 10 weeks in which two physical education lessons per week were scheduled; the lessons include specific themes for the development of the body schema and contain exercises to arouse pupils’ interest in sports and physical activities from this age. A number of 150 pupils aged between 5 years and 6 months and 7 years and 6 months participated in the experiment. The pupils were divided into two groups: the experimental group (76 pupils, 32 boys and 34 girls) and the control group (74 pupils, 31 boys and 33 girls). Next, the “Draw-a-Person” Test was applied at the beginning and end of the experiment. Results: The analysis of the results highlights the fact that after completing the intervention plan there is a significant improvement in the results of the experimental group on the Maturity Scale and implicitly on the three development scales: Head Scale, Body Schema Scale, and Clothing Scale. Among the three scales, the Body Schema Scale has the greatest influence on the Maturity Scale (r = 0.841). Conclusions: The preparatory class produces a connection between the kindergarten and the school; furthermore, the physical education lessons with an emphasis on the development of the body scheme contribute to the development of maturity and school preparation of the pupils.

## 1. Introduction

The body schema is a representation of a body for action [1,2,3,4,5]. The body schema is a bodily parameter that has a function in action planning and control, which can be, furthermore, descriptive and coercive [6]. The body schema is important because it shapes the sense of spatial dimension of physical things and also the relation of the body with this dimension. From a kinetic point of view, the body schema allows the body to move consciously and unconsciously [7,8], as well as the feeling of agency and being the subject of one’s own actions [9,10,11].

Motor learning can be defined as ‘a set of processes associated with practice or experience leading to relatively permanent changes in the capability for skilled movement’ [12] (p. 238). The body schema has an important role in the motor learning process as the body schema organizes bodily movements from within the body toward the environment; additionally, it also influences motor development. It is a stage in which body movements and positions become known through contact with various objects in the environment. As a result, the body schema is considered the image that each person has of their own body in the relationship between their body segments and the environment [13].

In the ontogenetic development process, the body goes through several stages: (a) the stage of the suffering body (“le corp subi” in French) from the first quarter of life, (b) the living body (“le corp vecu” in French) which is characterized by sensorimotor experiences and lasts until the age of 3 years, (c) the perceived body (“le corp percu” in French), in which perceptual experience predominates and which lasts between 3 and 6 years, and (d) the represented body, between 7 and 12 years, where the methodical experience of motor activity predominates [14,15,16,17].

Recent studies that have analyzed the importance of the development of the body schema are divided into several directions. An important direction is the development of the body schema through different sports and physical activities. Among these we list swimming [18], dance [19], yoga [20], and chess [21]. Another direction is that of different types of disabilities or deficiencies. These studies focus on the role of the body schema in psychomotor interventions in children with motor coordination developmental disorders [22,23,24], motor deficiencies [25], amputees [26], psychiatric disorders [27], and anorexia nervosa [28]. School physical education also addresses this topic, especially at the level of primary education [29,30,31].

In psychology, the term maturity means “the state of development necessary and sufficient to ensure the normal activity of a physical or mental function. The final stage of a growth process” [32] (p. 266). Readiness for school is a form of child development; it marks that level of development at which school-type activity can fully contribute to the further development of the child’s personality and is placed between the ages of 5 and 7. Readiness for school is also defined as “a child’s stage of development, necessary for successful attendance at primary education” [32] (p. 171). Physical education, in the preparatory class, is therefore essential in the process of psychomotor development of pupils as, it must be noted, the psychomotor development of pupils has a direct effect on school results [33,34]. At the age of 5, children can identify and name almost all parts of the head, torso, and limbs [35]. At the age of 6, the body plan can be mastered if specific work is performed. Given these factors, if the physical education lesson is structured appropriately, in addition to the psychomotor level, the pupils also develop on a cognitive and affective level [36,37].

In Romania, Physical Education and Sport was introduced in the primary cycle in 2011, when National Education Law No. 1 was enforced (and is currently still in force). This particular law mandates (in article 263, paragraph 7) that physical education lessons in primary classes are taught by teachers with higher specialized education. Prior to this law, physical education and sports lessons were taught by classroom teachers and until 2011, primary education consisted of 1 to 4 classes. The new law introduces another class, namely the preparatory class. The pupils are admitted to the preparatory class starting at the age of 5 years and 6 months until 6 years and 10 months. The role of this class is to assist the transition from kindergarten to school and to help facilitate integration of the children into the school institution. It must be noted, these pupils are still in the perceived body stage, the stage in which motor organization begins and is essential in the development and control of their motor activity. The introduction of this class, the lowering of the school age and the introduction of physical education and sports lessons with specialized teachers in a short time, led to the appearance of a school curriculum that was not adapted to the needs of the pupils included in this system. This may be because it was mainly achieved by adapting the curriculum for secondary school classes to primary education. However, as indicated above, the body schema, together with the other components of psychomotricity (eye-motor coordination, coordination of movements, laterality, spatio-temporal orientation, etc.), are important for the development of primary school pupils.

As such, the **aim of this study** is to investigate the influence of the structured physical education plan on the body schema development of preparatory class pupils by identifying, describing, and comparing the results of an experimental group in contrast with a control group.

**Hypothesis** **1** **(H1).***There are statistically significant differences regarding the results obtained on the three scales (Head Scale, Body Schema Scale, and Clothing Scale) between the experimental and the control group following the application of the intervention plan*.

**Hypothesis** **2** **(H2).***There are statistically significant differences regarding the results obtained on the Maturity Scale between the experimental group and the control group*.

**Hypothesis** **3** **(H3).***There are statistically significant differences between girls and boys from the experimental group on the three scales (Head Scale, Body Scheme Scale, and Clothing Scale) after the application of the intervention plan*.

## 2. Materials and Methods

### 2.1. Participants

This study was conducted on a group of 150 pupils between 5 years and 6 months and 7 years and 6 months (M = 6.43, SD = 0.52). These pupils were divided into two groups: the experimental group (76 pupils, 32 boys and 34 girls) and the control group (74 pupils, 31 boys and 33 girls).

This study was carried out in compliance with ethical principles (agreement no. 2/2021) and was approved by the Timis County School Inspectorate. The parents of the children involved in this study were informed about this experiment and gave their written consent for participation. The intervention in the physical education and sports lesson was carried out by the first author of the present study, who is a full-time teacher in the school where the experiment was applied (National College of Banat, Timisoara, Romania).

### 2.2. Design and Procedure

In order to test the importance of introducing topics that contain elements of body schema development in the physical education (PE) school curriculum, the experimental group followed a special intervention program. The physical education lessons took place between September and December 2021 according to the schedule (2 lessons per week, each lesson with a duration of 50 min) for a period of 10 weeks. The initial testing was carried out in both groups in September, which meant that the intervention plan was implemented in the experimental group in the last week of September. The pupils’ one-week vacation in November was also accounted for in the study. During the 20 lessons carried out in the experimental group, topics concerning the development of the body schema were introduced, while in the control group, the lessons were carried out according to the school program that did not include these topics. The final testing took place in December, in the last week of school, before the start of the winter vacation.

### 2.3. Intervention Plan

The topics included in the intervention plan were: knowledge and awareness of body parts, perception and reproduction of movements, positions of the body and its segments, positions of the body and its segments in the mirror and in relation to other objects, as well as knowledge and awareness of body parts through dynamic games and application courses (Table 1).

The physical exercises chosen within each theme aimed for the pupils to name the parts of the body and to associate these same parts of the body with the movements performed. They also targeted the location of the parts of the body by naming, touching and moving them, but also locating them in relation to themselves or to other colleagues or to other objects (including in the reflection of a mirror). Later, the pupils applied their knowledge in dynamic games in which their body parts were used, but also in interactive tracks in which they moved in different forms and on different objects using certain body parts selected by the teacher or of their choice.

### 2.4. Measurement

In order to measure the level of development of the pupils’ body schema, the “Draw-a-Person” test was used [24] both before and after the application of the intervention plan. This test was applied in class, both to the experimental group and to the control group. The test was applied collectively; each pupil received a set of 7 colored pencils (blue, green, red, yellow, purple, brown, and black), an eraser, and a sheet of A4 paper placed vertically. The pupils were given the following instruction: “On this sheet of paper you will draw a little man, as beautiful as you can. If you want you can also color it”. The pupils were left to freely choose the location of the drawing on the page, the size of the drawing, whether to change the page position, the gender of the drawn character, as well as the chosen colors or the refusal to use colors. No help or appreciation was given and no criticisms or suggestions were made. Undecided pupils were encouraged with formulas such as: “Very good, continue” and those who asked questions received the following answers: “Do as you wish/as you feel like”.

#### “Draw-a-Person” Test

This test included 51 items according to which the human drawing was rated [38]. The total obtained served as the basis for the establishment of a scale of development/maturity. For the establishment of this Maturity Scale, in addition to the 51 items, evolutionary signs were considered coming to a total of 70 items [39]. Three scales were analyzed: the Body Schema Scale (includes 33 items: trunk, legs, arms, etc.), the Head Scale (includes 23 items: head, eyes, mouth, nose, etc.), and the Clothing Scale (includes 14 items: shoes, belt, etc.). Each item present was rated with 1 point.

### 2.5. Statistical Analysis

The test scores were converted to raw scores. The total of the items presented in the child’s drawing, on each analyzed scale, was calculated. The ANCOVA statistical technique was used to analyze the results. It was a covariance analysis which helped us observe the differences between the experimental group and the control group. In ANCOVA, the pretest (covariate) is compared with the posttest (outcome variable) in order to remove variability.

## 3. Results

### 3.1. The Results Obtained at the “Draw-a-Person” Test

#### 3.1.1. Covariance Analysis for the Head Scale, the Body Scheme Scale, and the Clothing Scale in the Experimental Group and the Control Group

The results obtained on the Head Scale in the experimental group (EM = 11.79, SD = 4.30) compared to the control group (EM = 9.43, SD = 3.62) were statistically significant (F = 22.97, *p* < 0.05). The Body Schema Scale in the experimental group (EM = 19.46, SD = 6.53) showed differences compared to the control group (EM = 14.76, SD = 5.89), these differences being statistically significant (F = 43.10, *p* < 0.01). Additionally, on the Clothing Scale, the experimental group (EM = 5.11, SD = 3.17) registered differences compared to the control group (S = 4.00, SD = 2.90), these being significant as well (F = 12.77, *p* < 0.01). Following the application of the intervention plan, it was observed that there were significant differences on all three analyzed scales (Table 2).

#### 3.1.2. Covariance Analysis for the Maturity Scale in the Experimental Group and the Control Group

The differences between the experimental group (EM = 35.57, SD = 13.77) and the control group (EM = 28.71, SD = 11.92) were statistically significant (F = 24.104, *p* < 0.05) (Table 3). Therefore, the intervention plan contributed to the development of the maturity of the pupils in the experimental group.

#### 3.1.3. Covariance Analysis of the Relationship between Gender and Head Scale, Body Scheme Scale, and Clothing Scale

On the Body Schema Scale, the results recorded by girls (EM = 19.97, SD = 6.60) compared to boys (EM = 17.62, SD = 7.25) were statistically significant (F = 44.20, *p* < 0.01). On the Head Scale for girls (EM = 12.19, SD = 3.77) compared to that of boys (EM = 10.61, SD = 4.16), the differences were also statistically significant (F = 29.37, *p* < 0.01). On the Clothing Scale, it could be seen (Table 4) that girls registered statistically significant differences (EM = 5.14, SD = 3.20) (F = 38.17, *p* < 0.01) compared to boys (EM = 4.92, SD = 3.28).

#### 3.1.4. Covariance Analysis of the Relationship between Gender and the Maturity Scale Following the Application of the Intervention Plan

On the Maturity Scale, the results recorded by girls from the experimental group (EM = 11.35, SD = 2.56) compared to boys (EM = 10.57, SD = 2.92) were statistically significant (F = 39.31, *p* < 0.01) (Table 5). 

#### 3.1.5. Linear Regression Analysis of the Influence of the Body Schema Scale/Head Scale/Clothing Scale on the Maturity Scale

In order to observe the influence of the results recorded at the three scales on the Maturity Scale, the linear regression statistical technique was used (Table 6). The interpretation of the recorded results, as presented in Table 5, shows us that the Head Scale influenced the Maturity Scale in a percentage of 78.9%, the Body Scheme Scale in a percentage of 84.1% and the Clothing Scale in a percentage of 70.9%. The precision of the estimate could be observed; the intervals were relatively narrow between the lower limit (2.628/1.696, respectively 3.251) and the upper limit (3.109/1.953/4.007).

The fact that these results did not have a zero value showed us that the parameters of the regression equation to characterize the influence of the head scale on the maturity scale were suitable, which therefore means that the regression model is significant (Table 5).

## 4. Discussion

In this study, a quantitative analysis of drawing skills by boys and girls ages 5 to 7 was conducted through measuring scores in human figure drawing (Draw-a-Person Test) [38,39]. The main goal was to determine how the physical education lessons provided to the preparatory class influence the scores on the Draw-a-Person Test.

First, the results of this study highlighted the importance of physical education and sports lessons in improving the Maturity Scale scores of preparatory class pupils. The Body Schema scale was best developed with the help of specific exercises included in the intervention plan. This influenced the Maturity Scale more when compared to the Head Scale and the Clothing Scale. Other studies have also highlighted the importance of psychomotor education on the psychosocial maturation of students [40].

Second, the findings revealed the occurrence of sex differences in scores in the Draw-a-Person Test as they emerge from the Maturity Scale. The results of the study showed significant differences between girls and boys on all three scales taken separately. It was most clearly observed on the Head Scale and the Body Scheme Scale. These differences were in favor of girls, which corresponded with previous findings in the literature on sex differences in drawing [41,42,43]. There may be different reasons for why sex differences might exist in the human figure drawing, some studies have suggested that girls focus more on details whereas boys are more attentive to the contours of an object when engaged in a drawing activity [44,45]. Other studies have suggested that boys and girls have adopted different attitudes toward a drawing task, with girls being more conformist, more applied, and attentive to details due to their willingness to respond to the assumed expectations of the examiner [41]. Accordingly, girls’ higher inclination to add details in their human figure drawing would result in higher scores in the Draw-a-Person Test since this test mostly awarded points for the inclusion of specific details [46].

Our study makes the connection between the body schema, maturity, drawing and physical education in the preparatory class. Physical education and sports lessons can contribute to the development of the school maturity of pupils. We recommend that, for the preparatory class, the performance of physical education lessons with themes that develop the body scheme along with other components of psychomotricity should be implemented as they both hold an important role in preparing pupils for the learning process. Future studies will focus on the role of other psychomotricity components on the learning process. Among these components we will mention: space-temporal orientation, laterality, oculomotor coordination and others.

In terms of its novelty, this study highlights the importance of physical education lessons in the development of school maturity when approaching topics that develop the body schema. Consequently, the effects of physical exercises go beyond the physical sphere and enter the cognitive area, and the discipline of physical education and sports contributes to the development of primary school pupils.

## 5. Conclusions

The intervention program through specific exercises which focused on body schema had the expected effect. It can be seen from the pupils’ drawings that there are significant differences after the application of the intervention plan. At this age, physical exercise can contribute to the development of pupils’ maturity. It is important to emphasize that in using physical exercises, the body schema scale had the greatest influence on the maturity scale; this emphasizes the role and importance of physical exercise. Likewise, pupils’ drawings at this age express a wide range of matters. The physical education teacher can observe and use this information when planning activities, which will depend on the gender and level of physical and maturity development of the pupils.

## Figures and Tables

**Table 1 children-09-01369-t001:** The dynamic games used in the intervention plan.

No.	The Name of the Dynamic Game	Description and Aim of the Dynamic Game
1	Bouquets with body parts	Pupils move all over the field. At the whistle, they stop and, depending on the announced number, form the bouquets with the body parts indicated by the teacher.
2	The mirror in motion.	A pupil plays the role of the captain who must lead the group without telling them anything. They are seated facing the rest of their colleagues and then perform movements using their body parts. The rest of the students then move imitating the movements of their colleagues.
3	The square	Place 16 circles on the ground forming a square. The circles are of 2 colors (8 red and 8 yellow). Pupils must cross the square on the color indicated by the teacher using different parts of the body.
4	The imitator	Pupils reproduce the typical movements of swimmers, cyclists, soccer players, etc. They must identify the body parts used.
5	The ball	The pupils have to keep the balloon in the air with different parts of the body.
6	The body in motion	The objective is for the pupils to become aware of the movements that the different parts of the body can make. The head: what movements can be made? (Yes, no, right, left, etc.). Arms (bending, extending, rotating), etc.
7	The colors in motion	Sheets of different colors are placed on the ground. On the green, the pupils touch the ground with my head. On the yellow color, they dance. On the purple color, they perform 2 jumps, etc.

**Table 2 children-09-01369-t002:** ANCOVA test for the Head Scale, the Body Schema Scale, and the Clothing Scale following the application of the intervention plan.

Scale	Group						F	Partial η²
	Experimental (*n* = 76)			Control (*n* = 74)				
	EM	M	SD	EM	M	SD		
HS Post test	11.79	11.68	4.30	9.43	9.55	3.62	22.97 *	0.13
HS Pre							112	0.43
BSS Post test	19.46	19.57	6.53	14.76	14.66	5.89	43.10 **	0.227
BSS Pre							150	507
CS Post test	5.11	5.09	3.17	4.00	4.03	2.90	12.77 **	0.080
CS Pre							228	0.609

Notes: * *p* < 0.05, ** *p* < 0.01, EM = Estimated Media, M = Media, SD = Standard Deviation, HS = Head Scale, BSS = Body Schema Scale, CS = Clothes Scale.

**Table 3 children-09-01369-t003:** ANCOVA test for the Maturity Scale.

Scale	Group						F	Partial η²
	Experimental (*n* = 76)			Control (*n* = 74)				
	EM	M	SD	EM	M	SD		
MS Post test	35.57	35.46	13.77	28.71	28.83	11.92	24.104 *	0.144
MS Pre							188.97	0.562

Notes: * *p* < 0.05, EM = Estimated Media, M = Media, SD = Standard Deviation, MS = Maturity Scale.

**Table 4 children-09-01369-t004:** The ANCOVA test for the results recorded on the 3 scales according to Gender.

Scale	Group						F	Partial η²
	Feminine (*n* = 45)			Masculine (*n* = 31)				
	EM	M	SD	EM	M	SD		
BSS Post test	19.97	20.96	6.60	17.62	16.19	7.25	44.20 **	0.54
BSS Pre							71.25	0.49
HS Post test	12.19	12.91	3.77	10.61	9.58	4.16	29.37 **	0.044
HS Pre							38.88	0.34
CS Post test	5.14	5.80	3.20	4.92	3.97	3.28	38.17 **	0.51
CS Pre							65.37	0.47

Notes: ** *p* < 0.01, EM = Estimated Media, M = Media, SD = Standard Deviation BSS = Body Schema Scale, HS = Head Scale, CS = Clothes Scale.

**Table 5 children-09-01369-t005:** The results obtained on the Maturity Scale according to gender (experiment group).

Scale	Group						F	Partial η²
	Feminine (*n* = 45)			Masculine (*n* = 31)				
	EM	M	SD	EM	M	SD		
MA Post test	11.35	11.89	2.56	10.57	9.81	2.92	39.31 **	0.51
MA Pre							59.37	0.44

Notes: ** *p* < 0.01, EM = Estimated Media, M = Media, SD = Standard Deviation, MA = Mental Age.

**Table 6 children-09-01369-t006:** Linear regression.

MS				
	R Square	F	Interval Coefficients Inferior Limit	Superior Limit
HS	0.789	554.37 **	2.628	3.169
BSS	0.841	782.49 **	1.696	1.953
CS	0.709	359.23 **	3.251	4.007

Notes: ** *p* < 0.01, HS = Head Scale, BSS = Body Schema Scale, CS = Clothes Scale, MS = Maturity Scale.

## Data Availability

The data are available upon request from the corresponding author.

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
