# Peer review of "Relationship between Body Schema and Scholar Maturity: A Study from the National College of Banat in Timisoara, Romania"

_children, 2022, doi:10.3390/children9091369_

Round 1
Reviewer 1 Report
Review of the manuscript entitled: The Effects of Body Schema Developement on Scholar Maturity in Children from Preparatory Class in Physical Education Lessons The manuscript submitted is appropriate to the subject matter and scientific rigor. The authors raised a very current issue at work, which is not only interesting from a scientific but also a practical point of view. Some remarks improving the quality of future research. and suggested changes and comments to the submitted manuscript in order to improve the quality of the planned research and future publications below:
1. Would you please check and correct reference number:
a) Om reference [19} the children was from „The children all lived in urban or suburban settings in and around Glasgow, Scotland” why the authors write in this article „ In Romania” . would you please check your reference number[19] and correct it.
b) In reference [21] we can read:’ According to Gallahue (1982), children at 3 years of age are able to identify and name only the most important parts of the head, torso, and limbs, whereas at 4 years, children can identify and name a larger number of parts of the head, torso, and limbs, and at 5 years, children can identify and name almost all parts of the head, torso, and limbs. In a study by Simons and Dedroog (2009), TD children of 3–6 years of age were able to point to and name these same parts of the body.”In article autor write : „At the age of 5, children can name body parts in a mature way [21].”
2. In the Design and Procedure chapter, please describe examples of physical education lessons in the experimental group for example the dynamic games
3. Could you please correct the bibliography in accordance with the journal's guidelines and standardize it. Put the full the name of the journal or abbreviations, pages numbers, Doi numbers, volume. Would you please add also the access for the article. See my comments below.
3. I believe that the discussion is too short, we should develop it and add conclusions
Remaining comments in the article in the attachment.

Author Response
Answer to Reviewer 1
Following the reply received, the authors would like to thank you for your appreciation and comments, with which we have made considerable improvements to the study. In the hope that we have not omitted any details, we present below the changes made:
- All the answers are colored in green in the text.
a. We moved the proposition regarding the age of pupils who enter in preparatory class to the next paragraph, there where we discuss about the situation in Romania.
b. We corrected the proposition.
2. We added Tabel 1 with the dynamic games.
3. We corrected the Bibliography. Thank you for attention and patience!
4. We add conclusions.
5. We wrote ‘Draw a Person test” instead of ‘Draw a Man test” because the group was gender mixed. Generally, every pupil drew a person conform with their gender. For this reason, we considered more appropriate the word “Person” instead “Man” as in the Goodenough test.

Reviewer 2 Report
Thank you for the opportunity to read this article.
In my opinion the title of the paper is too long and do not emphasizes that this is a local study.
The abstract emphasizes the aim of the study and the intention is good, but the authors should apply also innovative methods of nudging pupils to become passionate about sports.
Not all the references are relevant for this study. Some other recent papers might be cited (2019-2022).
The authors choose the term students for pupils (5-7 years old). Why?
The introduction has to be short 2-3 paragraphs with general assumption regarding state of art. then present the research questions and aim.
The literature review section is missing. It has to describe in detail the state of art, other papers and case studies that applied this method. The authors has to describe in this section the main variables: BSS, HS, CS and maturity scale
Regarding methodology, the process of subject selection is clear, but I am not sure the sample is representative. Which is the statistical population?
It is very unclear for me why did the authors used ANCOVA. They could used One-way Anova without interaction (xls)=Two-way Anova or a paired sample T test.
ANOVA is a process of examining the difference among the means of multiple groups of data for homogeneity.
Two-way ANOVA: Used to determine how two factors affect a response variable, and to determine whether or not there is an interaction between the two factors on the response variable.
ANCOVA is a technique that remove the impact of one or more metric-scaled undesirable variable from dependent variable before undertaking research.
The authors have to specify the representativity of the sample. It is important to specify the statistical population, the probability and the error.
The authors do not present enough details to replicate the study.
The Tables and figures are relevant and clearly presented with correct labelling and appropriate units.
For some variables SD (standard deviations) are very high showing o heterogeneity of population and probably nonparametric tests will be more appropriate.
Overall, I consider that the research should be redesigned.
Author Response
Answer to Reviewer 2
Following the reply received, the authors would like to thank you for your evaluation and comments, with which we have made considerable improvements to the study. We present below the changes made, all are in green color:
- The introduction part has been improved by adding additional and interesting information related to body schema. These have been taken from 15 extra studies published in the last years. Unfortunately, we did not find articles to analyze the variables of our study (BSS, HS, CS) to present them in the introductory part.
- We changed the title of the article.
- We complete the abstract.
- We replaced “students” with “pupils”.
- Because we did this research in one school, all the pupils of the school involved in the preparatory class were part of the study, some in experimental group and some in control group.
- We used Ancova because it is a method to test the effect of a certain factor on the variable when the variant has been eliminated (kept under control). In our case, we followed the effect of the intervention plan on the body schema, analyzing the results from the final testing on each scale keeping the initial testing under control. According to the theory, Ancova has more power in the analysis, the results being more accurate. Indeed, Anova can also be used to compare the differences, but the results are more accurate with Ancova.

Reviewer 3 Report
It is interesting and necessary the evidence provided by your study that will help to improve the quality of preparatory class in Romania.
Author Response
Answer to Reviewer 3
Following the reply received, the authors would like to thank you for your evaluation and appreciation.

Round 2
Reviewer 2 Report
It is wired how the authors keep the initial testing under control....They might offer some explanations.
Author Response
Answer to Reviewer 2
Thank you for the evaluation and appreciation of our work. Regarding the Ancova, in an experimental design with one or more treatment and control groups, a commonly used covariate consists of scores on some pretest, measured using the same scale as that for the dependent variable (i.e., a posttest). In such an ANCOVA model, the question posed is: "Is there a difference in the mean posttest scores after adjusting for pretest scores?" The pre-program measure or pretest is sometimes also called a “covariate” because of the way it's used in the data analysis – we “covary” it with the outcome variable or posttest in order to remove variability.
